# Highly Efficient Refractive Index Sensor Based on a Dual-Side Polished SMS Fiber Enabled by Femtosecond Laser Writing

**DOI:** 10.3390/s23073651

**Published:** 2023-03-31

**Authors:** Jinke Li, Shiru Jiang, Stuart Aberdeen, Sang-Shin Lee

**Affiliations:** 1Department of Electronic Engineering, Kwangwoon University, Seoul 01897, Republic of Korea; 2Nano Device Application Center, Kwangwoon University, Seoul 01897, Republic of Korea

**Keywords:** fiber-optic RI sensors, high sensitivity, wide RI detection range, fs-laser writing, SMS fiber, multimode interference theory

## Abstract

Fiber-optic refractive index (RI) sensors based on wavelength-shift-based interrogation continue to present a challenge in achieving high sensitivity for a wide detection range. In this paper, we propose a sensor for determining the RI of liquids based on femtosecond laser (fs-laser) writing of a dual-side polished singlemode–multimode–singlemode (SMS) fiber. The proposed sensor can determine the RI value of a surrounding liquid by detecting the dip wavelength in the transmission spectrum of the light propagating through the sensing area. The high RI sensitivity is attributed to the increased interaction area established by the fs-laser, which creates hydrophilic surfaces and maintains the wide detection range of the SMS structure. The results of the wavelength-shift-based interrogation reveal that the fabricated device exhibited a high sensitivity of 161.40 nm per refractive index unit (RIU) over a wide RI detection range of 0.062 RIU. The proposed device has high processing accuracy and a simple manufacturing process. Hence, it has the potential to be used as a lab-on-fiber sensing platform in chemical and biotechnological applications.

## 1. Introduction

Refractive index (RI) sensors are extensively used in measurements of the refractive index (RI) and concentration of liquids or gases for various applications in industrial and bioscientific fields. Among these, fiber-optic sensors have been actively explored owing to their unparalleled advantages, such as a compact size, fast response, high resolution, robustness in harsh environments, and immunity to electromagnetic interference [1,2,3,4,5,6]. Thus far, several fiber-optic designs for RI sensing have been presented, including Mach–Zehnder interferometers (MZIs) [7], Fabry–Pérot interferometers [8], fiber Bragg gratings (FBGs) [9,10], intermodal interferometers [11], and multimode interferometers (MIs) [12,13,14,15,16,17]. Generally, the sensing mechanisms rely on either the variation in optical intensity or the shift in wavelength to underpin the operation of the RI sensor. RI detection range, sensitivity, resolution, mechanical strength, and fabrication quality are critical parameters for assessing the performance of RI sensors based on wavelength-shift-based interrogation. In particular, the RI detection range largely determines the operating environment selection of the device. High sensitivity over a wide detection range is vital in biosensing applications that require the detection of extremely small variations in RI. However, there is a trade-off between sensitivity and detection range due to the dynamic range limitation, design and operating parameters, and noise and signal processing [18,19]. Achieving high sensitivity over a wide range is still a challenge.

A singlemode–multimode–singlemode (SMS) fiber structure comprising two single-mode fibers (SMFs) and one multimode fiber (MMF) has gained significant attention owing to its potential in achieving a wide detection range [16,17,20]. This can be attributed to the periodic self-imaging over an ultra-long evanescent field interaction length that occurs in the MMF section, realized through multimode interference (MMI) [21]. However, standard SMS fiber structures generally have poor sensitivity because the cladding of the MMF prevents the interaction of its evanescent field with the surrounding environment. This results in a weak interaction between the transmitted light and surrounding media. Hence, quite a few fiber processing techniques have been utilized to modify the cladding of the MMF section [22,23,24]. Fiber polishing is a widely used technique for modifying SMS structures owing to its ability to remarkably improve their sensing performances [16,17,25,26]. During fiber polishing, increasing the size of the polished region for sensing can effectively improve the sensitivity. However, a larger sensing region incurs more modal dispersion, significantly reducing the detection range [27]. It is desirable to optimize the polishing technique to increase the sensitivity of the sensor while retaining a wide detection range. Conventional polishing techniques have been utilized to enable the side polishing of fiber structures, including wheel polishing [16,17] and CO_2_ laser polishing [26]. However, these techniques often produce fiber-optic RI sensors with diminished sensitivity, which is attributed to the unstable structural heating processes, low polished surface quality, and drawbacks in making structural changes in the micron-scale region [16,17,26]. Recently, the femtosecond laser (fs-laser) has been reported to be an outstanding tool for micromachining optical fibers with high flexibility, minimal mechanical and thermal damage, and precise machining accuracy [28,29]. Notably, fs-laser has been successfully used for improving the sensitivity of the fiber-optic MZI sensors [30]. This relies on the unique mechanism of fs-laser processing, which causes nonlinear effects of multiphoton absorption and avalanche ionization of irradiated materials, induced by ultrashort pulses [31].

In this study, an RI sensor incorporating a dual-side polished SMS fiber structure was developed to provide high sensitivity over a wide RI detection range. The MMF section in the middle was processed via fs-laser writing to serve as a sensing region, and the transmission characteristics of the sensor and the impact of the polished depth were primarily investigated. A one-step direct fs-laser writing process was applied to embody the proposed device. The hydrophilic surface processed using the fs-laser increased the interaction area, resulting in a sensor with much higher sensitivity. Moreover, the wide detection range attributed to the MMI over a broad interaction area was preserved. In response to different concentrations of sucrose solutions, a high sensitivity over a wide detection range was achieved. The results demonstrate the potential of the designed sensor for real-time sensing applications.

## 2. Proposed Fiber-Optic RI Sensor and Its Sensing Mechanism

### 2.1. Configuration of the Proposed Fiber-Optic RI Sensor

A dual-side polished SMS fiber structure was fabricated with the aid of fs-laser writing, as shown in Figure 1a. This structure comprised two outer SMFs, one input and one output, and a central dual-side polished MMF. The dual-side polished MMF section processed by fs-laser writing bridged the input and output SMF sections, where the two polished areas were parallel to the *y–z* plane. The core and cladding were 8.2 and 125 µm in diameter for the SMFs (Corning SMF-28e), respectively, with corresponding RIs of $n_{SMF\_co}$ = 1.4504 and $n_{SMF\_cl}$ = 1.4447 at a wavelength of 1550 nm. The length of the MMF (FG105LCA, Thorlabs) was 4 cm with $n_{MMF\_co}$ = 1.4667 and $n_{MMF\_cl}$ = 1.4447. The diameters of the core and cladding of the MMF were 105 and 125 µm, respectively. The length *L* of the polished area, or the sensing region, was 2 cm. This length was chosen in order to provide a sufficient interaction area with the surrounding medium for implementing a wide detection range. The diameter of the MMF after fs-laser writing was *D_p_* in the *x–y* cross-section. The RI value of the surrounding medium in the sensing region is denoted as $n_{sur}$. Initially, the light was irradiated from the input SMF into the MMF, where it subsequently interacted with the surrounding medium to be measured. Finally, the transmitted light was detected at the end of the output SMF, as shown in Figure 1b. The captured light was analyzed using an optical spectrometer based on the transmission intensity over a wavelength range of 1525–1555 nm. The dip wavelength ($\lambda_{dip}$) observed in the transmission spectra is relevant to the RI values of the liquid to be measured.

### 2.2. The Working Mechanism of the Proposed Sensor

The $\lambda_{dip}$ value in the transmitted spectra is indicative of the proposed RI sensor and can be attributed to the MMI in the MMF section. The MMI refers to the interaction of several guided modes that generally exist in MMFs and waveguides. Rsoft BeamPROP (Synopsys, Mountain View, CA, USA), a commercially available beam propagation tool, was employed to determine the light propagation and illustrate the MMI in more detail. Figure 2a shows the calculated light-field evolution of the MMI in the *x*–*z* cross-section of the proposed sensor. Many nonsymmetrical higher-order modes were excited, and there was interference between them in the polished MMF, where destructive interference caused some optical losses due to radiation. Additionally, a secondary coupling loss was observed when the light was coupled back to the output SMF [17]. Here, $\lambda_{dip}$ denotes the spectral position corresponding to the lowest transmittance of the transmitted spectra over a finite wavelength range, mainly resulting from a destructive interference in the polished MMF section. For instance, the $\lambda_{dip}$ value obtained with the lowest transmittance of −19.8 dB in the wavelength range of 1525–1550 nm was 1533.8 nm, as shown in Figure 2b. Figure 2c shows the optical electric field distribution for the three selected wavelengths (1528.8, 1533.8, and 1538.8 nm) corresponding to position A in Figure 2a, indicating that the light power varies with the wavelength owing to the MMI effect.

Here, we take two representative modes (i.e., *j*-th and *k*-th) into account to explain the $\lambda_{dip}$ shift in the transmission spectra, which is expressed by the following equation [16]:(1)$$\frac{2\pi }{\lambda_{dip}}\left(n_{j}^{eff}-n_{k}^{eff}\right)L=\left(2N+1\right)\pi ,\;\left(N=0,\;\pm 1,\;\pm 2,\ldots \right)$$

Here, $n_{j}^{eff}$ and $n_{k}^{eff}$ represent the effective RIs of the *j*-th and *k*-th modes of the MMF section, respectively. These modes are assumed to be two of the multiple modes contributing to the destructive interference in the side polished MMF of length *L*. The decrease in *D_p_*, namely the decrease in the residual thickness of the dual-side polished SMS fiber, leads to a decrease in the difference between $n_{j}^{eff}$ and $n_{k}^{eff}$ [16]. Consequently, $\lambda_{dip}$ shifts to shorter wavelengths to satisfy Equation (1). As $n_{sur}$ increases, the difference between $n_{j}^{eff}$ and $n_{k}^{eff}$ increases, resulting in a red shift of $\lambda_{dip}$. Figure 3a–c displays the calculated transmission spectra when the $n_{sur}$ value increases from 1.3325 to 1.3945. From these results, it can be confirmed that $\lambda_{dip}$ red-shifted with increasing $n_{sur}$ and blue-shifted with decreasing *D_p_*. Figure 3d shows the relationship between the wavelength shifts of $\lambda_{dip}$ and $n_{sur}$ for different values of *D_p_*. In Figure 3d, the observed wavelength shifts are 1.67, 2.93, and 6.24 nm at *D_p_* values of 91, 85, and 79 µm, respectively, when 1.3325 < $n_{sur}$ < 1.3945. The $\lambda_{dip}$ of the proposed structure is proportional to $n_{sur}$ but inversely proportional to *D_p_*. The calculated RI sensitivity was verified by fitting the simulation data, as shown in Figure 3d. The results exhibit a maximum sensitivity of 166.69 nm/RIU (with a coefficient of determination of *R*^2^ = 0.9631) at *D_p_* = 79 µm when 1.3571 < $n_{sur}$ < 1.3945. To achieve the highest sensitivity in the determined detection range, the fabricated *D_p_* was set to 79 µm.

## *3.* Fabrication and Experiments

### 3.1. Fabrication of the Proposed Sensing Device

The fabrication process consisted of two parts: fiber splicing and polishing. First, the coatings of the SMF and MMF sections were stripped off. Subsequently, the MMF section was spliced seamlessly into two sections of the input/output SMF using a splicer (FiberFox Mini 6S+, Englewood Cliffs, NJ, USA), as illustrated in Figure 4a,b. The outer diameters of the SMF and MMF were identical at 125 µm, while the core diameters were 8.3 μm and 105 μm, respectively. The length of the MMF section after fiber splicing was 4 cm as per the simulation setting. Second, for the polishing process, the spliced SMS fiber was clamped at either end by two fiber rolling stages (SFH-25L, ST1, Metropolitan, Republic of Korea) mounted on the monitored stages (XYCV630-C-N, Misumi, Tokyo, Japan), as shown in Figure 4c. Subsequently, a polished area was created on the SMS fiber with the aid of an fs-laser system (Lasernics, Daejeon, Republic of Korea), providing a linearly polarized collimated beam centered at a wavelength of 1040 nm, with a mean M^2^ of 1.3. The repetition rate and pulse duration of the generated laser pulses were 1 kHz and 250 fs, respectively. The fs-laser beam was delivered through a chain of turning mirrors and tightly focused on the surface of the SMS fiber via a dry lens (PL-L-40×, Lissview, Kuala Lumpur, Malaysia) with 40× magnification and a numerical aperture of 0.6. The fabrication procedures were monitored by a charge-coupled camera device connected to a computer. To fabricate the polished area on the SMS fiber, the laser beam was first focused on the central axis of the MMF cladding, 2 cm from the spliced interface of the SMF and the MMF. The focused beam was then moved along the *y* direction by 62.5 μm. When the mechanical shutter opened, the focused fs-laser beam scanned the fiber at a speed of 150 μm/s in a raster pattern in the *x–y* plane, as shown in Figure 4c. The laser-written lines were spaced 2.5 μm apart, and the scanning track covered 2 cm (*L* in the simulation) along the *x* direction, consisting of 8000 scanning lines. To ensure sufficient polishing, the width of the scanning track along the *y* direction was set to 125 μm. The mechanical shutter was closed when the scanning was completed. Next, the laser focus was moved 1 μm along the *z* direction to the central plane of the fiber, and the same scanning track was repeated to create the second layer. The distance between the laser focus scanning plane and the fiber center plane (at least 39.5 μm) was sufficiently secured to avoid spherical aberration during the laser scanning process [32]. This process created a polished area on one side of the fiber. To create a dual-side polished SMS fiber, the same method of scanning two layers was followed again after rotating the fiber rolling stages by 180° to create a polished region on the opposite side.

The hydrophilicity of a polished surface determines its wettability by water and indicates the adhesion between the surface and the solution. High hydrophilicity means that the polished surface has a large contact area with the external medium, which in turn leads to a better interaction between the propagating light and the external medium when compared to a smooth surface [33]. This results in a significant increase in sensitivity to RI variations [34]. Therefore, selecting an appropriate polishing technique is necessary to optimize the sensitivity of fiber-optic sensors. Conventional polishing methods, such as wheel polishing and CO_2_ laser polishing, have limitations in terms of their ability to enhance the hydrophilicity of fused silica surfaces and are therefore inadequate for this purpose. This is because their mechanisms restrict them from improving hydrophilicity solely by controlling surface roughness [35,36]. Meanwhile, direct fs-laser writing on the fused silica surface has the advantage of not only creating micro/nanostructures through surface modification but also being able to control surface roughness, resulting in enhanced hydrophilicity [35,36,37]. This approach is highly effective in reinforcing the sensitivity of fiber-optic sensors.

The surface quality of the polished region assisted by fs-laser writing with an average power of 12.0 mW was examined via scanning electron microscopy (SEM) and a 3D surface profiler (VK-X3000) as shown in Figure 5a,b, respectively. The images show that the polished surface had fluffy structures, and one of the micro/nanostructures was confirmed to be highly hydrophilic [37,38,39]. The ratio (*r*) between the actual and flat surface areas was measured as 1.462, indicating that the contact angle decreases according to the Wenzel model [33]. The decrease in the contact angle resulted in increased hydrophilicity. Moreover, the arithmetical mean height (Sa) and arithmetic mean roughness (Ra) were measured as 202 nm and 91 nm, respectively, further assisting in the enhancement of hydrophilicity [33]. The values of the arithmetical mean height (Sa) were 244, 214, and 202 nm in response to average pulsed powers of 8.9, 10.4, and 12.0 mW, respectively, indicating that the polished surface becomes smoother with stronger pulse powers. Similarly, the ratios (*r*) between the actual and flat surface areas were found to be 1.424, 1.450, and 1.462 for the same average pulse powers, suggesting that the hydrophilicity of the polished area increased with increasing laser power. To achieve a high hydrophilic surface and to minimize scattering losses caused by the rough polished surface [31], an average pulse power of 12.0 mW was utilized to embody the proposed dual-side polished SMS fiber, which is expected to deliver a high RI sensitivity. Figure 6a shows the fabricated SMS fiber structure with a polished length of 2 cm, with visible light used to illuminate the polished area. The practical value of *D_p_* was measured as 79 μm, which is consistent with the optimized value in the simulation, as shown in Figure 6b–d.

### 3.2. Results and Discussion

The device was first mounted on a vibration-isolated optical table. Subsequently, it was immersed in several sucrose solutions in deionized (DI) water at room temperature with $n_{sur}$ ranging from 1.3325 to 1.3945 to explore the liquid RI sensing characteristics of the prepared sensor. The RI range corresponded to the concentration of the sucrose solution (*C*), which was varied from 10% to 45%, in accordance with $n_{s}=0.033C^{3}+0.0456C^{2}+0.1416C+1.3166$ [40]. The room temperature was monitored in real time and maintained at 20 °C. Figure 7a shows the experimental setup for the RI measurement. The sensor was fixed between two fiber rolling stages with a glass base horizontally placed directly below the sensor where the solution was held. The input and output SMFs of the dual-side polished SMS fiber structure were connected to a light source with a wavelength range of 1520–1610 nm (OFB-AFB, LiComm, Gyeonggi-do, Republic of Korea) and an optical spectrometer (NIRQuest512-2.5, Ocean Optics, Orlando, FL, USA), respectively. A laboratory dropper was used to drop sucrose solutions to fully cover the dual-side polished SMS fiber structure. The sensor was meticulously cleaned and air-dried prior to each measurement with a sequential combination of DI water, ethanol, and DI water, thereby suppressing the influence of the residue solution. The evolution of the transmission spectra induced by the surrounding RI variation was then assessed after filtering out the noise signal.

Figure 7b shows one of the four experimentally measured results of the sensing device over an RI range of 1.3325 to 1.3945 (1.3325 < $n_{sur}$ < 1.3945). A clear red-wavelength shift in the transmission spectra was observed with increasing $n_{sur}$. This suggests that the device responded efficiently to the surrounding RI change in a wavelength-shifted manner. Specifically, $\lambda_{dip}$ underwent an average shift of approximately 6.13 nm, which was estimated from four measurement results with 1.3325 < $n_{sur}$ < 1.3945, as shown in Figure 7c. The standard deviation and repeatability limit of the $\lambda_{dip}$ shift of the four experimental results were calculated as 0.1226 and 0.4159 nm, respectively, when 1.3571 < $n_{sur}$ < 1.3945. This implies that a measured range of +/− 0.4159 nm from the mean value of 6.13 nm is within a 95% confidence interval. The maximum deviation of the $\lambda_{dip}$ shift among the four experimental results was 0.05, 0.20, 0.40, 0.25, 0.36, and 0.30 nm, corresponding to $n_{sur}$ values of 1.3444, 1.3571, 1.3657, 1.3740, 1.3847, and 1.3945 RIU, respectively. These acceptable deviations demonstrate the good repeatability of the device. The RI sensitivity of the proposed device was verified by fitting the experimental data, which is marked as a slope in Figure 7c. The device exhibited a higher sensitivity of 161.40 nm/RIU (*R*^2^ = 0.9999) when 1.3571 < $n_{sur}$ < 1.3945 compared to 28.06 nm/RIU (*R*^2^ = 0.9237) when 1.3325 < $n_{sur}$ < 1.3571. This can be attributed to the nonlinear response of the device to $n_{sur}$ [14,15]. When $n_{sur}$ approaches the RI of the multimode fiber (MMF) core, the critical angle for total reflection at the interface between the fiber core and the surrounding medium increased. Hence, light transmission through the interface and the interaction with the surrounding medium increased, substantially enhancing the sensitivity to variations in RI [16,17]. Therefore, the device exhibited a non-linear response to changes in the RI of the surrounding medium, with different sensitivities at distinct ranges of $n_{sur}$. Additionally, the measured maximum RI sensitivity was close to the expected 166.69 nm/RIU. This illustrates that the proposed sensing device suitably facilitates a good sensing performance owing to the hydrophilic polished surface processed by fs-laser writing. For practical applications, the RI of the target liquid can be determined by comparing the measured $\lambda_{dip}$ data shown in Figure 7c.

Table 1 presents comparisons between the various fiber-optic RI sensors based on SMS fiber structure in terms of their RI sensitivity and detection range. The results indicate that most of the devices were limited to a narrow RI detection range of less than 0.06 RIU and a low sensitivity with values of less than 160 nm/RIU, preventing their effective use in real-time monitoring in many environments [12,13,14,15,16]. In comparison, the proposed device achieved a high sensitivity of 161.40 nm/RIU over a wide RI detection range of 0.062 RIU. The wide detection range benefited from the periodic self-imaging transpiring in the ultra-long sensing region of the dual-side polished SMS structure. The fs-laser writing allowed the experimental sensitivity to fully match the calculated sensitivity.

Furthermore, a comparison of the sensitivity between side-polished SMS sensors fabricated using fs-laser writing and those fabricated using conventional polishing techniques confirms the improved sensing performance achieved with fs-laser writing [16,17]. For instance, dual-side structures polished using wheel polishing with similar polishing depths of 20 μm exhibit a limited sensitivity of only 85.29 nm/RIU [16], which is significantly lower than the RI sensitivity of 161.40 nm/RIU demonstrated in this work. The improvement in sensitivity can be attributed to the polished surface’s high hydrophilicity and minimal mechanical and thermal damage, resulting from the cold processing method of the fs-laser [29,30]. These findings demonstrate the potential of fs-laser writing to be used to enhance the sensing performance of side-polished fiber structures.

## 4. Conclusions

An RI sensor incorporating a dual-side polished SMS fiber structure based on fs-laser writing, whose operation is underpinned by the MMI effect pertaining to the MMF section in the middle, was demonstrated to exhibit high sensitivity over a wide detection range. The RI value of the target liquid surrounding the side-polished sensing region could be detected with respect to the position of $\lambda_{dip}$ in the obtained transmission spectra. The experimental results confirm that the manufactured device achieved a high sensitivity of 161.40 nm/RIU over a wide RI detection range of 0.062 RIU. The measured high sensitivity is attributed to the increase in the interaction area brought about by the fs-laser processed hydrophilic surfaces. The wide RI detection range was preserved, which is attributed to the periodic self-imaging transpiring in a broad sensing region in the MMF section of the dual-side polished SMS structure. The proposed sensor is expected to facilitate real-time monitoring of a variety of chemical and biological substances.

## Figures and Tables

**Figure 1 sensors-23-03651-f001:**
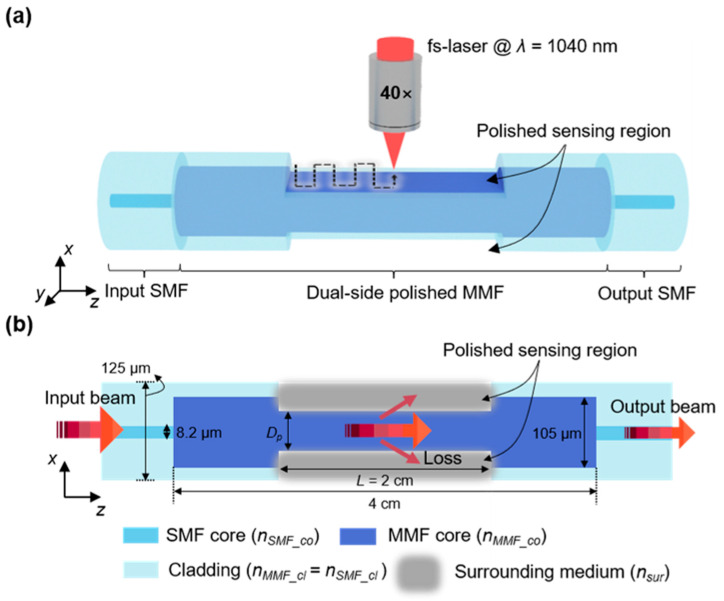
Schematic of the fabrication process for dual-side polished SMS fiber: (**a**) process of polishing the SMS fiber structure, enabled by the fs-laser writing. (**b**) Cross-section of the dual-side polished SMS fiber structure.

**Figure 2 sensors-23-03651-f002:**
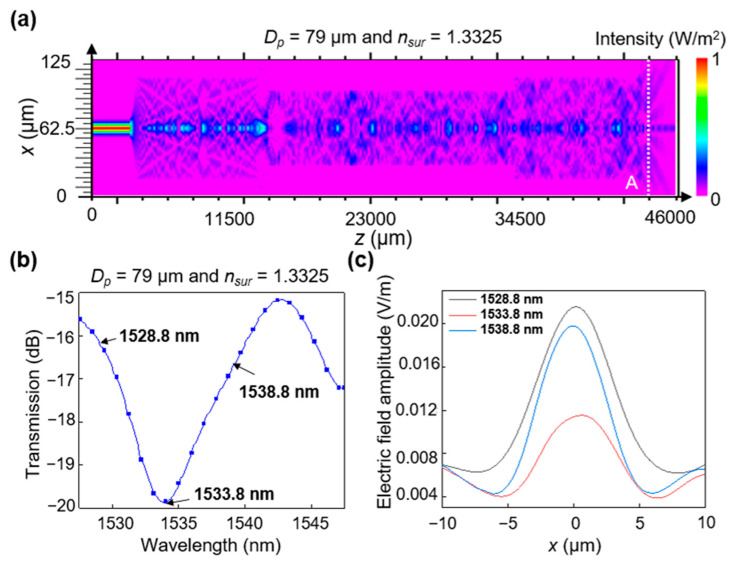
Calculated transmission characteristics of the sensor: (**a**) MMI behavior of the optical field propagating along the dual-side polished SMS fiber surrounded by the medium (*n_sur_* = 1.3325) at a wavelength of 1533.8 nm. (**b**) Calculated transmission spectrum of the proposed device surrounded by the medium (*n_sur_* = 1.3325) with *D_p_* = 79 µm. (**c**) Calculated optical electric field amplitude at the cross-section corresponding to position A (*z* = 43,000 µm) in (**a**).

**Figure 3 sensors-23-03651-f003:**
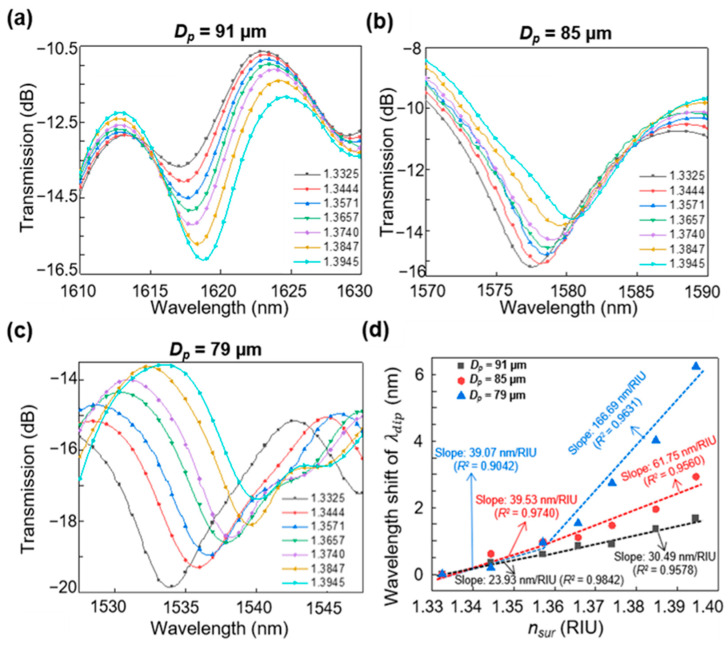
Calculated transmission spectrum evolution of the proposed sensor with *D_p_* values of (**a**) 91, (**b**) 85, and (**c**) 79 µm when 1.3325 < $n_{sur}$ < 1.3945. (**d**) Wavelength shifts of $\lambda_{dip}$ in response to $n_{sur}$ for *D_p_* = 91, 85, and 79 µm.

**Figure 4 sensors-23-03651-f004:**
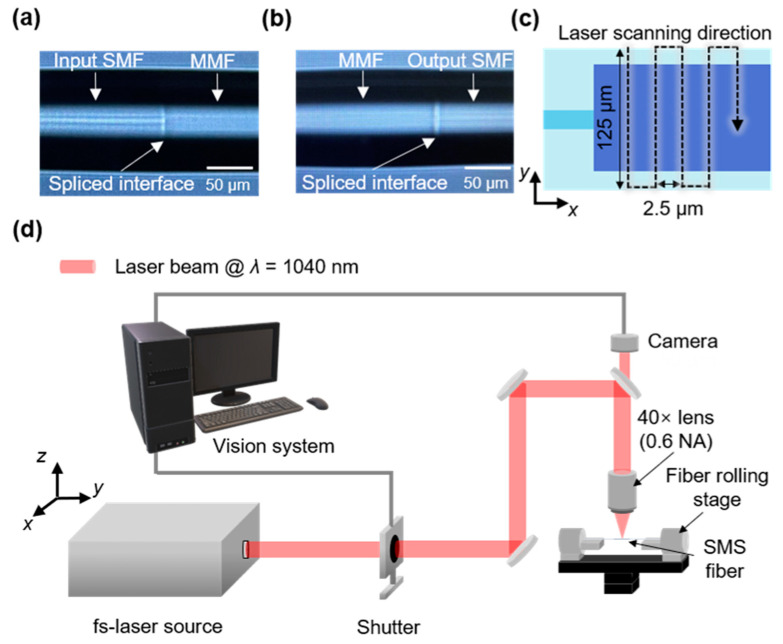
Optical microscope images of the splicing interfaces of the MMF: (**a**) input SMF and (**b**) output SMF. (**c**) Scanning track of the fs-laser for polishing the SMS fiber in the *x–y* plane. (**d**) Schematic of the fs-laser system used for the dual-side polished SMS fiber structure.

**Figure 5 sensors-23-03651-f005:**
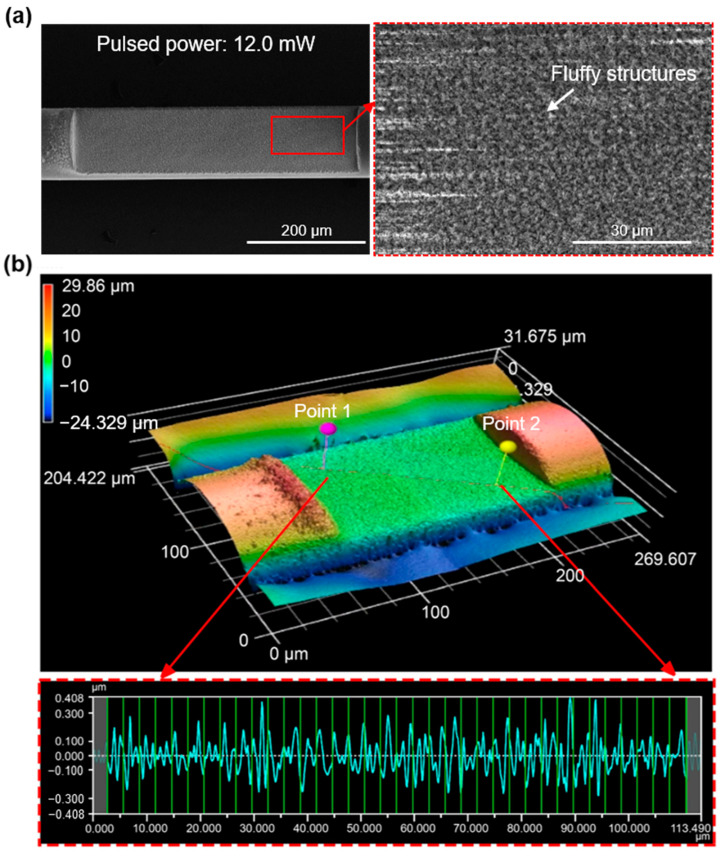
(**a**) SEM image and (**b**) 3D view of the polished area of the side-polished SMS fiber under a pulsed power of 12.0 mW corresponding to *L* = 500 μm and *L* = 200 μm, respectively.

**Figure 6 sensors-23-03651-f006:**
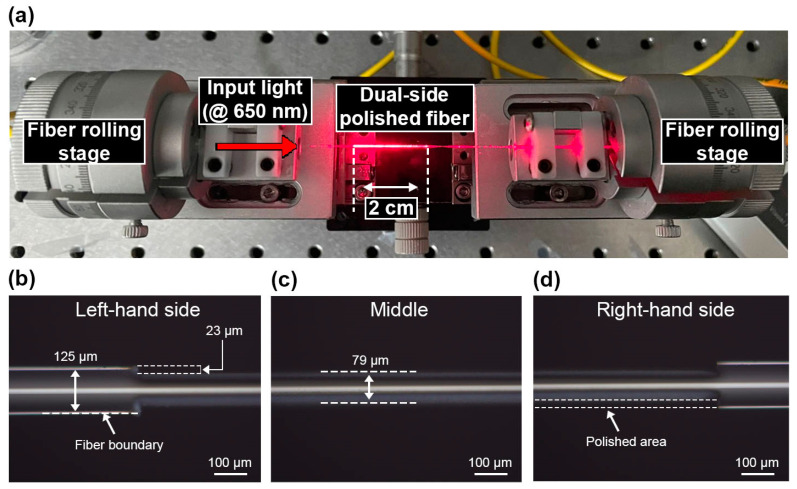
(**a**) Fabricated dual-side polished fiber fixed on two fiber rolling stages. Optical microscope images of the (**b**) left-hand, (**c**) middle, and (**d**) right-hand sides of the dual-side polished fiber under 10× magnification.

**Figure 7 sensors-23-03651-f007:**
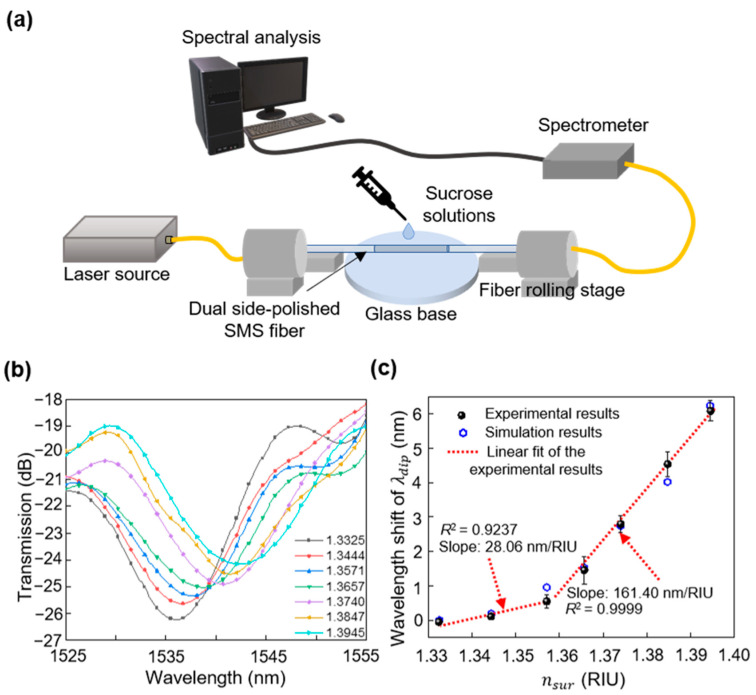
(**a**) Experimental setup of the fs-laser polished SMS fiber for sensing surrounding RI. (**b**) Measured transmitted spectra of the dual-side polished SMS fiber when 1.3325 < $n_{sur}$ < 1.3945. (**c**) Measured and calculated dependence of the wavelength shift of $\lambda_{dip}$ on $n_{sur}$.

**Table 1 sensors-23-03651-t001:** Comparison of fiber-optic RI sensors based on SMS fiber structure.

Sensor Structure	RI Detection Range (RIU)	Sensitivity (Maximum)	Reference
Half-tapered SMF–MMMF–half-tapered SMF	0.0180 (1.3320–1.3500)	68.12 nm/RIU (1.3320–1.3500)	[12]
SMF–MMF–SMF–MMF–SMF	0.0500 (1.3400–1.3900)	100.97 nm/RIU (1.3400–1.3900)	[13]
SMF–NCF–SMF–MMF–SMF	0.0480 (1.3330–1.3810)	113.66 nm/RIU (1.3330–1.3760)	[14]
SMF–MMF–NCF–SMF	0.0270 (1.3370–1.3640)	131.71 nm/RIU (1.3370–1.3640)	[15]
Tapered SHCS + side-polished	0.0600 (1.3450–1.4050)	151.29 nm/RIU (1.3450–1.4050)	[16]
Single-side polished SMF–MMF–SMF	0.0600 (1.3300–1.3900)	65.00 nm/RIU (1.3300–1.3900)	[17]
Dual-side polished SMF–MMF–SMF	0.0620 (1.3325–1.3945)	161.40 nm/RIU (1.3571–1.3945)	This work

NCF: no-core fiber; SHCS: singlemode–hollow core–singlemode; MMMF: multimode microfiber.

## Data Availability

Not applicable.

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
