# Peer review of "Highly Efficient Refractive Index Sensor Based on a Dual-Side Polished SMS Fiber Enabled by Femtosecond Laser Writing"

_sensors, 2023, doi:10.3390/s23073651_

Round 1
Reviewer 1 Report
In this article, a wide RI detection range and high sensitivity sensor is introduced. This sensor is based on femtosecond laser writing of a dual-side polished SMS fiber. The fabricated device demonstrated a high sensitivity of 161.40 nm per refractive index unit (RIU) over a wide RI detection range. However, there are some suggestions should be considered before publication.
1) In Figure 7 (c), the device exhibited a higher sensitivity of 161.40 nm/RIU when 1.3571 <n 1.3945 compared to 28.06 nm/RIU when 1.3325 < n < 1.3571. What causes the difference in sensitivity? It is suggested that authors elaborate on its mechanism.
2) In addition, authors' list of their own work in Table 1 may mislead readers, because the sensitivity of 161.40 nm/RIU is not measured in the full range (1.3325–1.3945), and it is suggested that the authors modify it.
3) In order to make the interface highly hydrophobic, the surface morphology should be prepared into fluffy structure. However, this will inevitably increase the residue of the solute to be measured. Will this affect the repeatability of the sensor? And will washing with alcohol affect the change of RI? It is suggested to increase the repeatability of sensor test.
4) Femtosecond laser processing is the most important step in this paper. It is suggested that the author give more detailed processing technology and parameters to enhance the readability of the document.
5) The paper shows that the morphologies of the polished surface are characterized by fluffy structures, because of its high hydrophilicity, washing with DI water before each measurement has some effect on the measurement. And will washing with alcohol affect the change of RI? In addition, it is recommended to increase the repeatability of the sensor test.
6) Generally, after the outer cladding of fiber is removed, it is easily affected by temperature and vibration. How can authors suppress the noise in the experiment?
Reviewer 2 Report
The article reported by Li et al. entitled “Highly efficient refractive index sensor based on a dual-side polished SMS fiber enabled by femtosecond laser writing” present the numerical and experimental validation of refractive index sensor using dual-side polished fiber made developed using Fs laser. The article reported a high sensitivity of 161.40 nm/RIU for a wide range of solution ranging from 1.3325 to 1.3945. All in all, this is a good piece of work and well supported with numerical and experimental validation. The article is a good match to the goal of Sensors; however, there are some concerns that need to be addressed:
1. What is the rule of optimization? how the optimized values are chosen a brief discussion is required?
2. The authors should describe the technique of fabrication of the sensor, a technical detail is must. How they avoid spherical aberration at interface?
3. In conclusion, authors are advised to discuss and outline the sensing performance of the device.
4. How many times the experiment was performed and what was the maximum deviation achieved from each trial. It is advised to show the error bar.
5. Presentation and technical merit of the work seem quite satisfactory for the reader but grammar and syntax need to be improved.
Round 2
Reviewer 1 Report
I agree that the paper is accepted.
Author Response
Thanks for your kind comments.
The attachment file is for the response to the academic editor.
If you are interested, take your time checking.
Best regards!